Positive association between epiphytes and competitiveness of the brown algal genus Lobophora against corals

Eich Andreas 1 2 eich.andreas@web.de
Ford Amanda K. 1 2 3
http://orcid.org/0000-0002-1399-9787 Nugues Maggy M. 4 5
McAndrews Ryan S. 1 2
Wild Christian 2
http://orcid.org/0000-0003-0930-5356 Ferse Sebastian C.A. 1 2
1 Department of Ecology, Leibniz Centre for Tropical Marine Research (ZMT) , Bremen , Germany
2 Department of Marine Ecology, FB2 Biology/Chemistry, University of Bremen , Bremen , Germany
3 Stockholm University, Stockholm Resilience Centre , Stockholm , Sweden
4 EPHE, PSL Research University, UPVD-CNRS, USR3278, CRIOBE , Perpignan , France
5 Labex Corail, CRIOBE , Moorea , French Polynesia
Pawlik Joseph
Electronic publication date: 2019 Feb 8
Publication date: 2019
Volume: 7
Electronic Location ID: e6380
Received 2018 Jun 15; Accepted 2018 Dec 28
Copyright: © 2019 Eich et al.
Copyright year: 2019
Copyright holder: Eich et al.
License: This is an open access article distributed under the terms of the Creative Commons Attribution License, which permits unrestricted use, distribution, reproduction and adaptation in any medium and for any purpose provided that it is properly attributed. For attribution, the original author(s), title, publication source (PeerJ) and either DOI or URL of the article must be cited.
License URL: https://creativecommons.org/licenses/by/4.0/

Keywords: Lobophora, Coral algae competition, Epiphytes, Allelopathy, Turf algae

Funding: (German) Federal Ministry of Education and Research (BMBF) through the ‘Nachwuchsgruppen Globaler Wandel 4 + 1’ (REPICORE) 01LN1303A This work was supported by the (German) Federal Ministry of Education and Research (BMBF) through the ‘Nachwuchsgruppen Globaler Wandel 4 + 1’ (REPICORE, grant number 01LN1303A). The funders had no role in study design, data collection and analysis, decision to publish, or preparation of the manuscript.

==============================
Observations of coral–algal competition can provide valuable information about the state of coral reef ecosystems. Here, we report contact rates and apparent competition states for six shallow lagoonal reefs in Fiji. A total of 81.4% of examined coral perimeters were found to be in contact with algae, with turf algae (54.7%) and macroalgae of the genus Lobophora (16.8%) representing the most frequently observed contacts. Turf algae competitiveness was low, with 21.8% of coral–turf contacts being won by the algae (i.e. overgrowth or bleaching of coral tissue). In contrast, Lobophora competitiveness against corals was high, with 62.5% of contacts being won by the alga. The presence of epiphytic algae on Lobophora was associated with significantly greater algal competitiveness against corals, with 75.8% and 21.1% of interactions recorded as algal wins in the presence and absence of epiphytes, respectively. Sedimentation rate, herbivorous fish biomass, and coral colony size did not have a significant effect on Lobophora–coral interactions. This research indicates a novel and important role of epiphytes in driving the outcome of coral–algal contacts.

Introduction

Coral reefs are rapidly degrading worldwide, with hard corals commonly being replaced by benthic algae (Diaz-Pulido et al., 2009). These changes are triggered by a combination of stressors, including overfishing of herbivores, excess nutrient and organic matter input, global climate change, and emergent marine diseases (Barott et al., 2012). To understand the mechanisms involved in these shifts, it is important to investigate interactions between corals and algae. These interactions typically involve a range of physical, microbial, and chemical mechanisms (McCook, Jompa & Diaz-Pulido, 2001; Barott & Rohwer, 2012). Algae can damage corals via direct physical interactions such as shading and abrasion, the exudation of allelopathic substances, and through indirect influences such as the release of dissolved organic carbon (DOC) (Barott & Rohwer, 2012). Microorganisms within coral mucus can profit from released DOC and deplete oxygen concentrations in proximity to coral tissue, which can subsequently cause mortality and disease (Barott & Rohwer, 2012; Jorissen et al., 2016).

Furthermore, the effects of algae on corals depend on the specific coral (Rasher et al., 2011) and algal (Jompa & McCook, 2003a; Bonaldo & Hay, 2014) species present. Crustose coralline algae generally have positive effects on coral reefs, for example, through promoting coral recruitment (Ritson-Williams et al., 2009) and suppressing macroalgal recruitment (Vermeij, Dailer & Smith, 2011), whereas turf algae can negatively influence coral health through impeding coral recruitment success (Birrell, McCook & Willis, 2005) and reducing tissue thickness (Quan-Young & Espinoza-Avalos, 2006). Anthropogenic influence (Barott et al., 2012), particularly sedimentation and terrestrial run-off, can increase the frequency of coral–turf contacts and turf competitiveness (Nugues & Roberts, 2003; Gowan, Tootell & Carpenter, 2014).

Brown macroalgae within the genus Lobophora have been a focus of recent research because of their rapid increase on many reefs worldwide (Mumby, Foster & Fahy, 2005). This genus can flourish following disturbances such as bleaching events (Diaz-Pulido et al., 2009) and storms (Roff et al., 2015), reaching benthic cover of up to 50% (Slattery & Lesser, 2014). Direct contact between Lobophora and corals commonly induces coral bleaching and mortality (Jompa & McCook, 2002; Slattery & Lesser, 2014; Vieira et al., 2016a) and results in decreased coral growth, reproduction, and recruitment (Nugues & Bak, 2006). Lobophora can also act as a substrate for a diverse epiphytic community (Fricke et al., 2011). However, few studies to date have considered the roles of epiphytes in coral–algal interactions or have distinguished the effects of epiphytes from those of their algal hosts.

Epiphytes have a wide range of effects on their host, and consequently their influence on host competitiveness is challenging to predict. Epiphytes can stress their host through shading (Round, 1981), can reduce consumption of host algae by herbivores (D’Antonio, 1985; Fong, Smith & Wartian, 2006; Smith et al., 2010), and if firmly attached can lead to tissue lesions and/or cause bacterial infections (Fricke et al., 2013). Some epiphytes are hemiparasitic and drain organic carbon from their host using penetrating rhizoids (Garbary & Deckert, 2002). If epiphytes act as a stressor to their algal host and reduce algal growth rates, they could potentially decrease the competitiveness of their algal hosts against corals. Alternatively, epiphytes could reinforce competitiveness of their algal hosts by directly damaging corals. For example, a common epiphyte of Lobophora, Anotrichum tenue (Fricke et al., 2011), can overgrow living coral tissue (Jompa & McCook, 2003b). Furthermore, by altering the composition and concentration of secondary metabolites in their host (Kremb et al., 2014), epiphytes can deter herbivorous fishes from consuming algal hosts (Karez, Engelbert & Sommer, 2000) and thus increase host competitiveness.

The goal of this observational study was to assess the types and outcomes of coral–algal interactions on coral reefs in Fiji and to gain first insights into the relationship between the presence of epiphytes and algal host competitiveness against corals. In addition to characterising the coral–algal interactions and recording the genera involved, we measured herbivorous fish biomass and sedimentation rates as explanatory variables. We expected a negative relationship between herbivorous fish biomass and the number of coral–algal interactions, and also anticipated that higher sedimentations rates would enhance the prevalence of turf algae involved in interactions by deterring herbivory (Bellwood & Fulton, 2008).

Material and Methods

Benthic surveys were conducted at two sets of three shallow lagoonal reef sites (four to six m depth) with increasing distance from each of two rural villages at Beqa Island (18°25′S, 178°08′E), Fiji (Fig. 1). At each reef, three 10 m transects were haphazardly placed on the benthos. For the first 10 hard coral colonies underneath each transect, we measured (to the nearest centimetre) the coral perimeter in contact with specific organisms (e.g. algal types, sponges, other corals) and bare substrate, thus additionally obtaining the total perimeter of the coral colony. For each observed coral–algal contact, we recorded the coral and algal genera, the presence of epiphytic algae, and the apparent outcome of the interaction. In this study, epiphytes were defined as all filamentous algae growing on the surface of macroalgae (Fig. 2) and turf algae as filamentous algae with a canopy height below two cm growing on abiotic substrate. A coral–algal interaction was considered as ‘alga-winning’ if the coral was bleached below or next to the interaction zone or if there was visible overgrowth of the alga over the coral surface, as ‘coral-winning’ if the coral was visibly overgrowing the algae, or as ‘neutral’ if neither was observed (Barott et al., 2012).

Figure 1 Beqa island in the south of Fiji (see arrow) with locations of the six study sites (stars) close to the village of Dakuibeqa and Dakuni.

Figure 2 Lobophora covered with filamentous epiphytes Scale ticks are in mm.

Photo by Andreas Eich.

At each site, three sediment traps, designed according to Storlazzi, Field & Bothner (2011), were deployed three times, each for ca. 3 weeks. Sediment material was filtered, dried for 24 h at 105 °C, and weighed. The abundance of herbivorous fishes (according to categorisations by Green & Bellwood, 2009) was assessed for six 30 × 5 m belt transects per site. Mobile and large fishes (>10 cm fork length) were counted in a first pass along the transect tape, and smaller, site-attached fishes were counted on the return pass. All surveys were performed by one surveyor (RSM) at approximately the same time of day (morning–noon), tide (outgoing), and weather conditions (calm) within 1 month (30.09.2015–29.10.2015). Fishes were categorised into five cm size classes (fork length) and the average fish biomass per site was calculated as described by Froese (2006), using species-specific length-weight conversion factors (Green & Bellwood, 2009).

For each coral colony, the proportion of the colony perimeter in contact with each type of contact category (i.e. algal genera, other benthic organisms, or bare substrate) was calculated (hereon referred to as ‘contact rate’). The two most frequently observed algal groups (i.e. turf and Lobophora) were analysed in more detail. For both types of algae, the relationship between contact rate and site was analysed with a generalised linear model. For contacts with turf algae, a generalised linear mixed model incorporating transect nested in site as a random effect was applied to analyse the relationships between contact rate and sedimentation rate, herbivorous fish biomass, coral genus, and colony size (perimeter). For contacts with Lobophora, we also tested the effect of the presence of epiphytes. Furthermore, models were run separately for each of the three most frequently observed coral genera. Following the same approach, the influence of the above parameters on the competitiveness of turf algae and Lobophora was tested for each coral genus incorporating a random factor for the coral colony ID. Competitiveness was defined as the proportion of the coral perimeter in contact with turf algae or Lobophora for which the interaction was classified as ‘alga-winning’. Since response values consisted of proportions, a beta distribution was chosen for all models (Kieschnick & McCullough, 2003). Proportional data contained the extremes (0 and 1) and was therefore transformed by (y (n − 1) + 0.5)/n, where n is the sample size (Smithson & Verkuilen, 2006). Comparisons of groups (i.e. tests comparing between different sites and presence or absence of epiphytes) were only conducted if at least two groups with at least three replicates each were present after sub-setting the dataframe for the separate analyses of coral genera. All models were reduced by removing the least significant term if a likelihood ratio test between the reduced and unreduced model revealed no significant difference. The effects of explanatory variables were tested using Wald tests. To account for multiple comparisons, p-values within individual analyses (i.e. of contact rate and competitiveness, respectively, both for turf algae and Lobophora) were adjusted after Holm (1979). All statistical analyses were carried out using R (Version 3.4.2).

Results

On average, 81.4% ± 2.7% (mean ± SE, n = 179 colonies) of coral perimeters and 156 of the 179 colonies examined were in contact with algae. Turf algae was the most common algal group (124 affected colonies) involved in coral–algal interactions, followed by the macroalga Lobophora (73 affected colonies), recorded as being in contact with 54.7% ± 3.2% and 16.8% ± 2.3% of coral perimeters, respectively (Fig. 3). All interactions involving turf algae or Lobophora were classified either as ‘alga-winning’ or ‘neutral’ (i.e. no ‘coral-winning’ interactions were observed).

Figure 3 Overview of observed coral–algae contacts.

(A) Mean percentage (± SE) of coral perimeter in contact with turf algae for which turf algae apparently won the interaction. (B) Proportions of coral perimeter in contact with different algae: Turf algae (‘turf’, green), Lobophora (‘Lobo’, brown), other phototrophic organisms like macroalgae (e.g. Padina or Halimeda) or benthic cyanobacteria (‘other phototrophic, light grey), benthic invertebrates like sponges or corals (‘other benthic’, dark grey), and abiotic substrate like sand, rubble, or recently dead coral skeleton (‘abiotic’, white), (C) Mean percentage (± SE) of coral perimeter in contact with Lobophora with (‘yes’) and without epiphytes (‘no’) for which Lobophora apparently won the interaction.

Contact rates with turf algae increased significantly with sedimentation, while herbivorous fish biomass, coral genus, colony size, and site had no significant effect (Table 1A). Separate analyses for the three most frequent coral genera in contact with turf algae (Porites n = 67 of 179 contacts, Pavona n = 16, and Acropora n = 12) revealed no significant effect for any of the explanatory variables and species (Table 1A). None of the explanatory variables were found to have a significant effect on turf algal competitiveness when considering all coral types (21.8% ± 3.4% ‘alga-winning’ contacts, Fig. 3A), or when analyses were run separately for the three most common coral genera (Table 1A). The effect of site could not be analysed for Acropora corals, since after sub-setting the data, only one site remained with at least three replicates. The contact rates of Lobophora were significantly higher when epiphytes were present (Table 1B). Sedimentation rate, herbivorous fish biomass, coral genus, site, and colony size did not influence Lobophora contact rate. When analysing contact rates of the three most frequent coral genera individually (Porites n = 44, Acropora n = 8, and equally frequent Favites and Pavona n = 6), no significant effects were found (Table 1B). Most interactions (62.5% ± 5.2%) between Lobophora and corals were classified as ‘alga-winning’ (Fig. 3C), and in total 12.4% ± 2.1% of the perimeter of all corals was negatively affected by Lobophora (i.e. ‘alga-winning’, Fig. 3C). Site, sedimentation rate, herbivorous fish biomass, coral genus and colony size (perimeter) did not significantly influence Lobophora competitiveness. However, the presence of epiphytes had a highly significant (positive) effect on the proportion of ‘alga-winning’ Lobophora–coral interactions (Fig. 3C; Table 1B). Consistent with the results of the overall model, the presence of epiphytes increased ‘alga-winning’ rates for Porites–Lobophora contacts. No significant effects of the explanatory variables were found for the other coral genera (Table 1B). For Acropora and Pavona, the effects of epiphytes and site could not be analysed due to insufficient replication after sub-setting of the data. Likewise, the effect of site could not be tested for Pavona corals.

Table 1 Effects of sedimentation (SED), herbivorous fish biomass (HFB), coral perimeter (CP), presence of epiphytes (EPI), site, and genus (GEN) on interactions of coral colonies with turf (A) and Lobophora (B).

(A) Turf	
	Main effect	
	n	SED	HFB	CP	EPI	SITE	GEN	
Contact rates	
Overall	124	χ2 (df = 1) = 9.948, p < 0.05	n.s.	n.s.	–	n.s.	n.s.	
Coral genus	
Porites	67	n.s.	n.s.	n.s.	–	n.s.	–	
Pavona	16	n.s.	n.s.	n.s.	–	n.s.	–	
Acropora	12	n.s.	n.s.	n.s.	–	n.s.	–	
Competitiveness	
Overall	124	n.s.	n.s.	n.s.	–	n.s.	n.s.	
Coral genus	
Porites	67	n.s.	n.s.	n.s.	–	n.s.	–	
Pavona	16	n.s.	n.s.	n.s.	–	n.s.	–	
Acropora	12	n.s.	n.s.	n.s.	–	n.t.	–	
(B) Lobophora	
	Main effect	
	n	SED	HFB	CP	EPI	SITE	GEN	
Contact rates	
Overall	73	n.s.	n.s.	n.s.	χ2 (df = 1) = 8.399, p < 0.05	n.s.	n.s.	
Coral genus	
Porites	44	n.s.	n.s.	n.s.	n.s.	n.s.	–	
Acropora	8	n.s.	n.s.	n.s.	n.s.	n.s.	–	
Favites	6	n.s.	n.s.	n.s.	n.s.	n.s.	–	
Pavona	6	n.s.	n.s.	n.s.	n.s.	n.s.	–	
Competitiveness	
Overall	73	n.s.	n.s.	n.s.	χ2 (df = 1) = 16.024, p < 0.005	n.s.	n.s.	
Coral genus	
Porites	44	n.s.	n.s.	n.s.	χ2 (df = 1) = 13.770, p < 0.0005	n.s.	–	
Acropora	8	n.s.	n.s.	n.s.	n.t.	n.t.	–	
Favites	6	n.s.	n.s.	n.s.	n.s.	n.t.	–	
Pavona	6	n.s.	n.s.	n.s.	n.t.	n.t.	–	
Notes:

Separate analyses were run for total percentage of contact (‘contact rates’, above) and for percentage of contact for which the interaction was apparently won by the alga (‘competitiveness’, below). Results are shown for GLM analyses of overall coral contacts with turf and Lobophora, and for GLMM analyses run separately for the 3 and 4 main coral genera in contact with algae, respectively. p-Values were adjusted for multiple comparisons after Holm (1979).

n.s., not significant; n.t., not tested because number of groups <2 or number of replicates within groups <3.

Discussion

This study investigated the contact rates and apparent outcomes of interactions between corals and algae in a lagoonal reef system in Fiji. Contrary to our expectations, herbivorous fish biomass had no influence on coral–algal interactions. However, as expected, we found a positive correlation between the rates of sedimentation and of coral–turf algae contact. This observation was also made previously in Saint-Lucia, Caribbean (Nugues & Roberts, 2003), and in Moorea, French Polynesia (Gowan, Tootell & Carpenter, 2014), and could be explained by deterred grazing, which was not measured directly and does not necessarily have to be reflected in herbivorous fish biomass. This influence was not observed when coral genera were analysed separately, probably because replication was reduced due to sub-setting of the data.

We found a negative influence of Lobophora on the coral perimeter, which was approximately twice as high as previously reported for Lobophora variegata in the Caribbean island of Curaçao (12.4% ± 2.1% vs. maximally 5.7% of negatively influenced coral perimeter, Nugues & Bak, 2006). Epiphytes significantly increased the proportion of interactions in which Lobophora were reported to be winning. We assume that this overall effect on the investigated corals was mainly caused by the high proportion of Porites–Lobophora contacts, which were strongly affected by the presence of epiphytes. Other coral genera (Acropora, Favites, and Pavona) were far less common, which resulted in low replication and high variance when running the model separately for these genera. The number of Pavona corals observed was so low that solely interactions with Lobophora overgrown by epiphytes were found (i.e. no interactions were recorded between Pavona and Lobophora without epiphytes present), precluding statistical analysis of this factor. Generally, results for these less common genera should be interpreted with caution.

The enhanced competitiveness when epiphytes were present on Lobophora could be driven by direct impacts of the epiphytes themselves on neighbouring corals and/or via indirect effects on the algal host. In New Caledonia, bleaching of coral perimeters in contact with L. herderacea has been proposed to be caused by associated epiphytic filamentous algae (Vieira, Payri & De Clerck, 2015). Functionally, epiphytes are similar to turf algae, which can damage corals (McCook, Jompa & Diaz-Pulido, 2001; Quan-Young & Espinoza-Avalos, 2006). In this study, however, turf algae were associated with a relatively higher proportion of neutral interactions, suggesting that epiphytic filamentous algae and filamentous turf algae growing on abiotic substrate differ in their competitiveness, potentially due to differences in their species composition.

Indirect effects of epiphytic algae on coral–algal interactions could occur via chemical alterations of the algal host. For example, extracts of Lobophora overgrown by epiphytes have a higher activity against human immunodeficiency virus than extracts from Lobophora without epiphytes (Kremb et al., 2014). As Lobophora can damage corals via different allelochemicals (Rasher & Hay, 2010; Rasher et al., 2011; Slattery & Lesser, 2014; Vieira et al., 2016b), increased production or concentration of any of these chemicals in the presence of epiphytes could significantly influence host competitiveness in coral–algal interactions. Another potential explanation for the elevated competitive potential of Lobophora with epiphytes is that epiphytes deter grazing on algal hosts (D’Antonio, 1985; Fong, Smith & Wartian, 2006; Smith et al., 2010). Competition with corals can induce increased algal production of allelopathic chemicals at the expense of the production of anti-herbivore substances, which can result in higher algal palatability (Rasher & Hay, 2014). If epiphytes release Lobophora from grazing pressure, they could further facilitate production of allelochemicals that enhance algal competitiveness without the usual associated trade-off of increased palatability. Other parameters such as the duration of the coral–Lobophora contact or the species composition of the epiphytal community may also have had an influence on the outcome of the competitions. Due to infrastructure limitations stemming from a different initial scope of the study and the difficulty of accurately identifying filamentous algal species in the field, it was not possible to determine the species of epiphytes and turf algae involved in the interactions or perform an analysis of algal extracts for potential allelochemical compounds. This constitutes an important avenue for future studies of the role of epiphytes in affecting coral–algal interactions and the underlying mechanisms.

Conclusions

These results provide strong indications that the competitiveness of Lobophora against corals, particularly those within the genus Porites, is enhanced when epiphytes are growing on Lobophora. This study suggests a need to consider associated epiphytic algal communities when investigating coral–algal interactions. Although this study was limited in its scope and was unable to characterise the epiphytic assemblage or the chemical compounds involved in the Lobophora–coral contacts, we hope that future studies will extend on our findings to better understand the dynamics of coral–algal interactions. Further research could additionally experimentally manipulate epiphytes to determine whether this observation is a result of direct negative effects of epiphytic algae on corals or of an increased competitiveness of Lobophora and should investigate potential differences in algal communities growing on dead corals vs. on macroalgae.

Supplemental Information

Supplemental Information 1 Coral-algae contact rates raw data.

Click here for additional data file.

Supplemental Information 2 Herbivorous fish biomass raw data.

Click here for additional data file.

Supplemental Information 3 Sedimentation rates raw data.

Click here for additional data file.

We would like to thank the village elders of Dakuibeqa for research permission, and Manassa and Bose for their help during fieldwork. Furthermore, we want to thank the ZMT dive team for their support. Comments by two anonymous reviewers helped to improve the manuscript. The research reported in this paper contributes to the Programme on Ecosystem Change and Society (www.pecs-science.org).

Additional Information and Declarations

Competing Interests

Author Contributions

Data Availability

The authors declare that they have no competing interests.

Andreas Eich conceived and designed the experiments, performed the experiments, analysed the data, prepared figures and/or tables, authored or reviewed drafts of the paper, approved the final draft.

Amanda K. Ford conceived and designed the experiments, performed the experiments, authored or reviewed drafts of the paper, approved the final draft.

Maggy M. Nugues conceived and designed the experiments, authored or reviewed drafts of the paper, approved the final draft.

Ryan S. McAndrews performed the experiments, authored or reviewed drafts of the paper, approved the final draft.

Christian Wild conceived and designed the experiments, contributed reagents/materials/analysis tools, authored or reviewed drafts of the paper, approved the final draft.

Sebastian C.A. Ferse conceived and designed the experiments, contributed reagents/materials/analysis tools, authored or reviewed drafts of the paper, approved the final draft.

The following information was supplied regarding data availability:

All raw data are available in Supplemental Files.

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
