# Peer review of "Positive association between epiphytes and competitiveness of the brown algal genus Lobophora against corals"

_PeerJ, doi:10.7717/peerj.6380_

## Round 0.1 · original submission · Major Revisions

I now have two substantial reviews from experts in this area of research. Both recommend revision of this manuscript, with Reviewer 2 asking for greater context in the paper and for more analyses of the reported data. Please respond to both reviewers' comments in a point-by-point fashion. I am likely to return the revision to the reviewers for further consideration.

Reviewer 1 ·

Basic reporting

Coral reefs are increasingly being replaced by other functional constituents, including fleshy macroalgae. However, as noted by the co-authors, the mechanisms of this competitive interaction has rarely been described. Here they test the competitive nature of Lobophora, as well as turf algae, with and without an epiphytic community. They demonstrate that: 1) turf algae doesn’t compete with corals despite a significant number of interactions, 2) Lobophora is a competitive “winner”, and 3) that competition is directly related to ephiphytic overgrowth (but not to sedimentation, herbivorous fish biomass, or coral size/species). The authors should be acknowledged for this interesting set of field-based observations that add to our understanding of algal-coral interactions, and raise many questions for (future) research.

The manuscript is reasonably well written in the context of Peer J style. The background is relatively inclusive of the existing relevant literature (but see below), and the provided data support the conclusions presented.

Experimental design

The approach is well described and generally analyzed, and the research question is timely and meaningful. The project seems quite preliminary (i.e., note vs article), but it doesn’t “oversell” the results obtained, and argues for further study.

Validity of the findings

The results are generally robust, and the conclusions are certainly intriguing. My primary concerns with this study are the lack of detail relative to the central questions/results. Specifically, the data demonstrate that epiphytes are the primary contributor to competitive dominance in Lobophora-coral interactions. I am disappointed that there wasn’t any attempt to describe the epiphytic community (ie, micro-algae, fouling inverts, biofilms, etc), and to identify key/any of these taxa using traditional and/or molecular approaches. Likewise, the authors note numbers of interations between turf and corals, and Lobophora and corals, but “corals” is a broad category in Fiji. Who specifically? And did you see more/less fouling on specific genera/species of coral? These seem like fundamental data for the paper as presented.

Secondarily, and maybe less important for a field study, I assumed that this paper would get to some mechanistic explanation for the results obtained (based on the structure and context of the introduction). I was therefore disappointed when I got to the end and realized this paper falls short, despite some discussion of chemical defenses, etc. To that extent, when the authors wrap around to their discussion of allelopathy [line 155: Lobophora can damage corals…], they should include references by Slattery & Lesser (2014), as well as Rasher et al (2010, 2011), that not only demonstrated allelopathy, but identified specific compounds responsible.

Reviewer 2 ·

Basic reporting

Overall, the manuscript is well-written but still has some sentences and paragraphs that could be improved, as pointed throughout this review.

A major problem is that the authors have much more to explore in their data. The presented results are too basic and represent an initial analysis. I would really like to see a more thorough exploration of the data they have collected (available in the supplementary material).

In the introduction: reading of the paragraph within lines 48-52 is disconnected. Part of it may be due to awkward writing, but I would suggest authors to work on a better connection between this paragraph in the intro.

The manuscript also lacks important references of the field of coral-seaweed competition, several of which were conducted in Fiji by Mark Hay´s group. A few examples are provided below:

Rasher, D. B., & Hay, M. E. (2010). Chemically rich seaweeds poison corals when not controlled by herbivores. Proceedings of the National Academy of Sciences, 107(21), 9683-9688.

Rasher, D. B., Stout, E. P., Engel, S., Kubanek, J., & Hay, M. E. (2011). Macroalgal terpenes function as allelopathic agents against reef corals. Proceedings of the National Academy of Sciences, 201108628.

Rasher, D. B., & Hay, M. E. (2014). Competition induces allelopathy but suppresses growth and anti-herbivore defence in a chemically rich seaweed. Proceedings of the Royal Society of London B: Biological Sciences, 281(1777), 20132615.

Rasher, D. B., Hoey, A. S., & Hay, M. E. (2017). Cascading predator effects in a Fijian coral reef ecosystem. Scientific reports, 7(1), 15684.

Clements, C. S., Rasher, D. B., Hoey, A. S., Bonito, V. E., & Hay, M. E. (2018). Spatial and temporal limits of coral-macroalgal competition: the negative impacts of macroalgal density, proximity, and history of contact. Marine Ecology Progress Series, 586, 11-20.

Bonaldo, R. M., & Hay, M. E. (2014). Seaweed-coral interactions: variance in seaweed allelopathy, coral susceptibility, and potential effects on coral resilience. PLoS One, 9(1), e85786.

Dixson, D. L., Abrego, D., & Hay, M. E. (2014). Chemically mediated behavior of recruiting corals and fishes: a tipping point that may limit reef recovery. Science, 345(6199), 892-897.

The authors argue the findings are essentially novel, but I think some of what they present have also been pointed out by the papers below. This does not reduce the importance of the manuscript or their findings, but by putting their results in the context of previous findings, authors will be able to highlight the contribution of their work.

Fong, P., Smith, T. B., & Wartian, M. J. (2006). Epiphytic cyanobacteria maintain shifts to macroalgal dominance on coral reefs following ENSO disturbance. Ecology, 87(5), 1162-1168.

Smith, T. B., Fong, P., Kennison, R., & Smith, J. (2010). Spatial refuges and associational defenses promote harmful blooms of the alga Caulerpa sertularioides onto coral reefs. Oecologia, 164(4), 1039-1048.

Lines 38-40, for example, the authors say: “However, the extent to which algae drive these shifts by outcompeting adult corals and the mechanisms influencing the outcomes of coral-algal competition are still unclear”

This is not true. There is an extensive body of literature on how algae drive these shifts and on the competition mechanisms as well. Therefore, I would suggest the authors to rethink/rewrite this statement.
Lines 48-49: awkward phrasing, please clarify.

Line 54: Nugues & Bak 2008 should probably be:

Nugues, M. M. & Bak, R. P. Differential competitive abilities between Caribbean coral species and a brown alga: a year of experiments and a long-term perspective. Mar. Ecol. Prog. Ser. 315, 75–86 (2006).

Please confirm that.

I could not find specific hypothesis for the influence of sedimentation and herbivore biomass on the frequency and outcomes of contacts. I suggest the authors to include this explicitly by the end of the intro.

Figure 4 seems overcomplicated to me. I think pie charts could do the work better. Also, Figure 4 and 5 could be easily combined into one unique figure to be more informative and straight to the point.

Experimental design

Overall, methods are well-described, but would benefit from some clarifications as described in the comments below:

There is no mention to which coral species the authors investigated or if they did it for all the coral species. There is enough evidence in the literature that the extent and result of seaweed-coral contact is strongly dependent on the identity of both interacting species (coral and seaweed).

Line 82: please replace randomly by haphazardly, which would probably describe your method better. If the authors decide to keep randomly in the text, the I would like to see a better description on how they made sure transects were random indeed.

Lines 83-84: this sentence can be simplified, for example: “we measured the coral perimeter to the nearest centimeter”.

Line 98: please provide more details on the tide conditions and the time frame of the study. For example, all fieldwork was conducted within a month, two months, an year? Please clarify.

Lines 115-116: awkward writing. Please review it. Also, you probably do not need to state that your showing means and standard error at this point of the text.

Line 124: authors provide a X2 result, which is not clearly described in the methods. Please improve the methods section where you describe the statistical analyses.

Lines 131-132: there is a bit of discussion here, please save it to the proper section of the paper.

Validity of the findings

The data is somewhat simple and provide a descriptive overview of coral-seaweed contacts. A major problem of the data is that the authors do not specify the coral species, which would enable a much more refined analysis. This would be of interest to a broader audience. Looking through their raw data, I observed they have this information. Therefore, I strongly suggest the authors to reanalyze the data using coral species, with specific tests for each species. This would strengthen the findings by showing how general the demonstrated outcome is for several coral species. Analysis, findings and conclusions presented so far are too simplistic, which limits the impact of the manuscript. A more profound literature review will also enhance the manuscript.

---

## Round 0.2 · Minor Revisions

Reviewer 2 was unable to provide further input, but Reviewer 1 has returned a second review and reports that the authors have provided satisfactory responses to most of the Reviewers' comments. Please note the complaints about the quality of the writing, which must be addressed in further revision, along with a point-by-point response to Reviewer 1's re-review.

Reviewer 1 ·

Basic reporting

The co-authors have done a reasonable job of addressing my comments (and those of the other reviewer). In many cases this included adding references and clarifying introductory/discussion points. On the whole the manuscript is improved, and meets the basic reporting requirements of Peer J. I will note however, that the added text also has many new typos/grammar issues. This suggests: 1) a rush job, likely not reviewed by all co-authors, and 2) a lack of respect for the time of reviewers/editors, and ultimately readers. Both points of which detract from an otherwise interesting story.

For example, line 286 of the track changed revision states: Lobophora to produce RATHER allelochemicals than... Pretty sure the correct word is OTHER but I can piece together some real sentences with "rather" and a couple of deleted words. It is not my job, or the readers, to try to figure out what you mean to say. CORRECT ALL, and all co-authors MUST proof before resubmitting!!!

Experimental design

As noted prior, this paper lacks some fundamental data (i.e. who are the epiphytes), and what is the mechanism of action; but the co-authors have noted that this is part of an MSc thesis from a remote location, implying that data will not be added here. They have apparently further analyzed some of the specific coral data, and avoided others due to a lack of replication. Unfortunate that more wasn't available, but there is still a story here that will certainly inform future algal/coral competition research.

Validity of the findings

I think the article provided is an important note that will direct future research efforts. My major complaint with the current manuscript is that the co-authors don't "own" their short-comings. Yes, they have made the case to their reviewers, and the editor, that we can't hold this project to the highest standards and/or expect additional data given circumstances. And, yes, there is still intriguing data here that should be put out to inform future studies. But it would be great if the next iteration noted these issues in the article. For example, the co-authors could state something to the effect of: Although this study wasn't able to identify the epiphytes involved (and/or substitute chemistry) for XXX reasons, future studies might approach this topic to better understand algal/coral competitive interactions...

Additional comments

A final revision needs to be carefully edited/proofed, and contextualized if you want this paper to become anything more than a throwawa (i.e., a cited product that defines future research efforts).

---

## Round 0.3 · accepted · Accept

The authors have done a good job of correcting the manuscript text, and I hope they will bear in mind that this should be done in advance of their original submission for subsequent submissions.

#